# Divine Action and Dramatic Christology: A Rereading of Raymund Schwager's *Jesus in the Drama of Salvation*

## Willibald Sandler

Department of Systematic Theology, University of Innsbruck, 6020 Innsbruck, Austria;
willibald.sandler@uibk.ac.at

**Abstract:** This article shows that and how Raymund Schwager's five-act dramatic Christology is at the same time a theology of divine action that takes a Christological, personal, dialogical and dramatic approach. Secondly, this article develops a methodological approach of *Jesus in the Drama of Salvation*, in which Schwager draws on Anselm of Canterbury and thus understands theology as an unfinishable project of "conversion of thought". This methodology is developed further in the form of a continuous self-application of theological insights to one's own theology, so that Schwager's dramatic five-act theology of divine action opens up to ever new readings. In this way, thirdly, Schwager's dramatic theology of the Gospels is developed in the direction of a biblically based dramatic-kairological phenomenology of divine action. According to this, God acts through and in Jesus by means of events that make the kingdom of God present to people in an exemplary way, thus pulling them out of previous entanglements of catastrophe and placing them in situations of new beginnings. This liberating action of God towards salvation does not overwhelm the free choice of human beings, but places them in Kairoi, i.e., in extraordinary times of salvation, in which they are released to actively accept this offer of salvation in favour of a salvific self-transformation, or to reject God in an aggravated way. On the methodological path described above, this kairological salvific action of God, mediated by Jesus' proclamation of the kingdom of God and by the work of the Holy Spirit, is grounded in God's new creative action of the incarnation of Jesus Christ. In this way there emerges an expanded sphere for God's personal and dramatic kairological salvific action, which embraces the whole of creation.

**Keywords:** divine action; biblical hermeneutics; spiritual theology; dramatic theology; theological methodology; phenomenology of the event; kairos; resurrection; new creation; kingdom of God

## 1. Introduction

The question of divine action is a controversial topic not only in academic theology, but it is also central to the Christian faith. Depending on how people believe in God, they have different ideas about what is possible for God and what can be expected of him. During the COVID-19 pandemic, the question of how God could allow such things to happen was conspicuously absent from the churches and the media.[1] Apparently, many Christians have "learned" that it is immature to blame God for something he cannot change. This corresponds to an increasingly widespread "semideistic" (von Stosch 2006, p. 102) "depotency strategy" of theodicy since the Second World War and the atrocities of the Shoah (von Stosch 2018, pp. 26–28): in the face of overflowing evil, people try to save faith in God's goodness by curtailing faith in his omnipotence.

It is astonishing that the biblical understanding of God took an exactly opposite development in the face of continued catastrophic shocks. Since the catastrophe of the Babylonian exile, the biblical writings of the Old Testament have not relativised God's power to save, but have increasingly emphasised it more strongly and comprehensively.[2] In the New Testament, Jesus proclaimed the kingdom of God as already dawning in the midst of this long-suffering world, and with his resurrection he let the new creation begin.

Furthermore, he encouraged people to turn to God in all their needs and to trust him with everything: "Your faith has saved you" (Mk 10:52)[3].

With regard to faith in God's power to save, a gap is opening up between modern skepticism and biblical confidence. This divide is also polarising the churches. While many Christians in the mainstream churches hardly expect any help from God in their concrete problems and therefore turn to him with little supplication, charismatic-evangelical Christians from renewal movements and free churches proclaim a God who can remove every need. In doing so, they refer to the faith that Jesus proclaimed in the Gospels. At the same time, they are not free of exaggerations and misunderstandings. Especially neo-charismatic movements nurture the expectation that God will solve all our problems, work miracles and create prosperity if only one believes and prays correctly (Sandler 2021, pp. 172–78). Such misunderstandings and abuses, in turn, strengthen the criticism of those Christians who consider a belief in God who saves from concrete hardships to be fundamentally problematic.

Here, theology has an important task to clarify how we can develop a strong faith in God's saving power on a biblical basis, which is also responsible and accordingly sustainable. In this way, it could help to overcome the polarisation between mainstream churches and charismatic-evangelical movements, offering the latter a better biblical hermeneutic and opening up to the former a strong, biblically grounded and accountable faith in a God who is "mighty to save"[4], not only for the hereafter but also in the face of present challenges and threats.

This requires a theology of God's action with special qualities: it would have to be biblically grounded (1), as well as theologically coherent and problem-sensitive (2), in order to arrive at life-relevant, vivid conclusions without simplification (3), which are spiritually fruitful in that they open up perspectives of meaning and hope even in times of individual and global crisis (4).

Many years of experience as a dogmatic theologian of the "Innsbruck Dramatic Theology" (Wandinger 2016) with Raymund Schwager's dramatic theology in professional theology and adult education, as well as with charismatic-evangelical renewal movements (Sandler 2021), have strengthened my conviction that Schwager's dramatic theology has high potential for a theology of God's action that meets these four criteria (cf. Sandler 2012). In this paper, I will present Schwager's most important book, *Jesus in the Drama of Salvation* (Schwager [1989] 1999), with its dramatic five-act structure of a divine action through Jesus Christ, as a biblical Christology and soteriology that is also a theology of God's action. In addition, I will show how it is guided by a method—inspired by an original interpretation of Anselm of Canterbury—that has hardly been considered in the reception of Schwager so far. From there, an approach to a dramatic theology of God's action emerges in the manner of a project open to further development.

The task of this essay is to present and develop this open approach to a dramatic theology of God's action. Its relationship to similar approaches and other theologies of God's action, as well as its application to the aforementioned tensions between modern skepticism and biblical confidence, will be the subject of a subsequent essay.

## 2. God's Action and Dramatic Christology in the Work of Raymund Schwager

A central concern of Raymund Schwager was the systematic theological critique of ideas of a violent God. With René Girard, he took as his point of departure the anthropological dynamics of mimesis (as the involuntary imitation of the desires of others) and scapegoating mechanisms that lead people to project their own collective violence onto God, thus generating "sacral" ideas of a God who is at once a blessing and a violent God (Palaver 2020). With Girard, Schwager was concerned to expose such processes of projection and to uncover an authentically Christian understanding of God, according to which God is all love and completely free of all violence. In doing so, he responded to the modern shift in the Christian image of God from a punishing, judging, and wrathfully unpredictable God to a God who is all love, without, however, erasing or trivialising the problem of violence.

*2.1. Revelation and God's Action in Must There Be Scapegoats? (1978)*

In *Must There be Scapegoats*? (Schwager [1978] 1987), Schwager tested Girard's theory of mimesis, scapegoating and sacralisation in terms of a coherent interpretation of the Bible. For a biblical hermeneutics, Schwager took over from Girard the guiding assumption of a deep tendency, insurmountable by purely human means, to "sacralise" images of God in the sense of projecting one's own collective violence onto images of God (Schwager [1978] 1987, pp. 1–42).

In light of this dynamic of original sin (Schwager [1997] 2006), Schwager understands with Girard "the Old Testament as a long and laborious exodus out of the world of violence and sacred projections" (Schwager [1978] 1987, p. 43). Accordingly, in *Must There be Scapegoats?* Schwager focused on the many biblical passages that describe God as acting violently. He divided these biblical texts into four categories, depending on whether: (1) God attacks people irrationally,[5] (2) he acts directly but justly, especially against apostasy or social injustice, (3) God exercises violence against people only indirectly by setting peoples against each other, or whether (4) there arises only an impression of divine violence created by the fact that people who have lost sight of God themselves turn against each other and commit violence in the name of God; thus, God is not an actor here, but withdraws or allows himself to be driven out by people (Schwager [1978] 1987, pp. 112–32).

Schwager noted a certain development in the Old Testament understanding of God and revelation towards the fourth category, but not in a continuous way without setbacks. Thus, he found in the Bible a history of God's self-revelation that calls and enables people to gradually overcome deeply rooted ideas of God that are obsessed with violence, as Girard has opened up with his concept of a scapegoat mechanism. According to Schwager, however, this process only comes to a conclusion in the New Testament, since there are different and contradictory lines of interpretation at the level of the Old Testament, where without the deepening perspectives of the New Testament it is not obvious which of them is leading and how they can be seen together (Schwager [1978] 1987, p. 115).

So, Schwager's synopsis of the two testaments shows: God's violence expressed in the Bible is "always [...] human power interpreted as God's action" (ibid., p. 63). In reality, "God's actions pursue a reign of justice, peace, and knowledge of God" (ibid., p. 115), and "God's judging action uncovers the mechanism of violence and leads humans to a totally different goal" (ibid., p. 107), namely, a new gathering with comprehensive reconciliation. God's non-violence corresponds to a dialogical action that not only respects the freedom of human beings, but also "is meant to spur humans on to new deeds of their own" (ibid., p. 167).

*2.2. Jesus in the Drama of Salvation as a Theology of God's Saving Work through JESUS Christ*

Schwager's major work, *Jesus in the Drama of Salvation* (Schwager [1989] 1999), is a biblical doctrine of salvation that structures the Christ event as recorded in the Gospels into a five-act drama (ibid., pp. 29–158):

Act 1: Jesus' proclamation of the kingdom of God;
Act 2: Jesus' words of judgement;
Act 3: Jesus' death or self-sacrifice on the cross;
Act 4: Jesus' resurrection;
Act 5: The sending of the Holy Spirit with the church planting effect of a new assembly.

In a first approximation one can read *Jesus in the Drama of Salvation* as a New Testament continuation of his twelve-years-older book *Must There be Scapegoats*? (Schwager [1989] 1999, p. ix). In this comparative perspective, Schwager's systematic approach to biblical texts with the distinction of seemingly contradictory text groups that require a coherent interpretation can be found again in the five acts of the drama of salvation. *Must There be Scapegoats?* dealt with the contradictions in the Old Testament between a God who is essentially described as merciful and compassionate (ex., 34:6; Ps 86:15, etc.), but who is nevertheless repeatedly portrayed as wrathful and violent. *Jesus in the Drama of Salvation*

dealt with seemingly incompatible statements about God's action through Jesus Christ in the Gospels.

This applies to the texts in the Gospels about the incipient kingdom of God, which Schwager assigns to the first act. In them, Jesus speaks to people of a new divine act of salvation without demanding a preceding conversion and by means of healings and acts of deliverance, Jesus conveys the experience that this new divine action has already begun (cf. Lk 11:20). These events of the coming kingdom of God are obviously in contrast to the words of judgement (second act) in which Jesus warns people of an impending hell if they do not repent. Because Jesus claims that God acts through him, a contrast emerges in Jesus' proclamation that once again calls into question the coherence of God's action. Other statements in the Gospels refer to Jesus' self-giving on the cross as a condition of salvation. These texts, which Schwager places in the third act, also seem to contradict Jesus' unconditional proclamation of the kingdom of God in the first act.

Schwager did not construct these two contradictions—between the first and second act and between the first and third act—solely from a systematising external view of the Gospels. Rather, in his studies on the history of soteriology (Schwager [1986] 2015), he was able to locate the contradiction between an "eschatological soteriology" that emphasises God's causeless goodness and a soteriology of the cross that underlines the weight of sin and God's justice as a continuous fundamental problem in the history of theology (Niewiadomski [1989] 2015, pp. 24–25). Further, he demonstrated both contrasts as a problem indicator in the contemporary discussions of German-language historical-critical exegesis. Accordingly, Heinz Schürmann—with recourse to Rudolf Bultmann and a sharpened reception by Peter Fiedler—tended, in view of the above tensions, to favour Jesus' proclamation of the kingdom of God (Act 1) as an authentically Jesuan "eschatological soteriology" and to relativise his words of judgement as well as a soteriology focused on the cross as a later church theology (Schwager [1989] 1999, pp. 10, 45–50, 56–57).

In contrast, Schwager's explicit concern was to develop a dramatic hermeneutic that would bring together these different types of statement into a coherent plot: by rejecting the offer of salvation proclaimed by Jesus in the name of God, the religious authorities placed themselves and their people in a situation of deepening desolation. This action of the people created a new situation (Schwager 2004), which Schwager assigns to a second act. Jesus responded to this with a changed proclamation of words of judgement instead of the previous announcement of salvation. Schwager did not interpret Jesus' words of judgement as an announcement of a direct punitive and violent judicial action of God, but as a warning of the hopeless consequences of this rejection, which the religious authorities would bring upon themselves and their people in the manner of a self-judgment. Thus, Jesus' original message of a totally good, loving God is confirmed and not compromised by the words of judgement (Schwager [1989] 1999). Since Jesus' words of warning belong to a different constellation of action and in this sense to a different act of the drama, they are no longer in irreconcilable opposition to his initial unconditional announcement of salvation. Rather, they clarify the necessity of God's offer of grace for salvation: this offer "does not presuppose conversion, but wants to awaken it" (ibid., p. 129). Thus it is no longer necessary to deny Jesus' words of judgement and to attribute them to a later church theology as opposed to Jesus' original intention.

The third act, which culminates in Jesus' death on the cross, is triggered by increasingly aggressive human actions, especially by the religious authorities of Israel, who did not heed Jesus' warnings, but were further provoked by them to the point of handing him over to be killed in the mistaken belief that they were carrying out God's judgement on a blasphemer. Thus, for Schwager, Jesus' death on the cross is above all the consequence of a blinded, violent reaction of Jesus' opponents on his proclamation of the kingdom of God. At the same time, however, it is an expression of Jesus' freely chosen willingness to allow himself to be drawn into the violent and blinded world of sinners in order to then break it open from within (ibid., pp. 117, 135, 144). In this sense, Jesus is also an active agent in the third act.

On the other hand, Schwager firmly rejects the basic soteriological assumption of Martin Luther, Karl Barth, and Hans Urs von Balthasar that Jesus' death on the cross is primarily an action of the Father on the Son is decisively rejected by Schwager (ibid., pp. 162–63). Admittedly, the Bible speaks—only in Paul, not in the Gospels—of the Father's handing over ("paradidónai") of the Son, but according to Schwager, this is only for the salvation of sinners through a redemptive breaking of their sinful entanglement. It is only for this purpose that the heavenly Father affirms the death on the cross as the redeeming gift of the Son. The murderous action of the sinners against Jesus is not directly intended by the Divine Father, but only accepted for a higher purpose (ibid., pp. 163, 165).

The fourth act is primarily determined by God's new action in raising Jesus from the dead. According to Schwager, this divine act is at the same time an act of judgement, in which God judges in favour of Jesus in the dispute between Jesus and his opponents over who could rightfully act on God's authority. However, because Jesus gave himself up for the people so that they would not be condemned, this judgement "is therefore, when considered more deeply, also a verdict in favor of sinners" (ibid., p. 135). After the people had rejected the forgiveness that Jesus implicitly promised them with his unconditional new offer of salvation, there was now a "redoubling of that readiness to forgive" (ibid., p. 136) by the Risen One. He offers this, first of all, to his disciples who, according to Schwager, were the first to become "personally and subjectively [...] guilty at his [Jesus'] downfall", because, in contrast to the people, they had, during his earthly ministry, gained a deeper insight into their master's mystery and yet, in the decisive hour, they deserted and betrayed him" (Schwager 2004, p. 352).

Schwager, together with the Evangelist Luke, distinguishes the work of the Holy Spirit from the work of the Risen One and assigns it to a separate fifth act. He justifies this step, which has been questioned by exegetes such as Gerhard Lohfink (Niewiadomski [1989] 2015, p. 26), with a new "pneumatic experience transforming the inmost self" (Schwager [1989] 1999, p. 143) of the disciples, which led to spontaneous praise of God and candid speaking in public. Thus, in the fifth act, a new action of God is revealed through the Holy Spirit. The Holy Spirit brings about an inner experience which is at the same time communal and leads to an open witness in public. In this way, the Holy Spirit brings about a new gathering of people in unity and without exclusionary opposition, which then gives rise to the Church. For Schwager, this work of the Spirit is the fruit of Jesus' redemptive self-giving:

> "Because the crucified one let himself be drawn into the dark world of his adversaries, far from God, and there lived out his obedience to the Father, the deep godless realms of the human heart themselves became the place where the divine spirit can from now on reach and touch people." (Schwager [1989] 1999, p. 144)

*2.3. Schwager's Dispute with Historical-Critical Exegesis in View of God's Action through and in Jesus Christ*

What was also new in *Jesus in the Drama of Salvation* compared to Schwager's earlier biblical-theological work was his consistent engagement with contemporary historical-critical exegesis. He problematised its methodological limitations in accordance with an Enlightenment understanding of history, according to which it is only prepared to accept human actors and not God's action to explain historical developments (Schwager [1989] 1999, pp. ix, 15–17, 30, passim). Biblical theologians who work exclusively in a historical-critical way would indeed confirm that Jesus *claimed* that God acted in him in a new and definitive way. However, they would not have the methodological means to seriously consider the truth of this claim and would therefore explain it exclusively in terms of the history of ideas, without getting involved in the question of truth, but what if, for historical reasons, the assumption that Jesus' claim is true turns out to be indispensable?

Despite the problem of a theologically unacceptable methodological narrowing of an exegesis that is not prepared to go beyond the historical-critical method, Schwager did not want to simply leave it aside for he was concerned with historical relevance and thus with historical proof of the truth of God's action in Jesus Christ. Therefore, he tried to open a

middle way between the historical and the dogmatic method in theology, which (Troeltsch 1922) had irreconcilably opposed a hundred years ago (Schwager [1989] 1999, pp. 119–20; Gmainer-Pranzl 2014, pp. 53, 48–60). Schwager did not want to simply take Jesus' claim to be the "instrument of a new and final act of God" (Schwager 1997a, p. 5, my translation) as true and draw conclusions from it that are not comprehensible for a historical method committed to Troeltsch's criteria of correlation and analogy. However, neither was he inclined to simply drop this claim of Jesus, which was of crucial importance. Instead, he placed it at the centre in the manner of a hypothesis, so that the essential results of his investigations into the message of the kingdom of God, the judgement and the cross—i.e., the first to the third act—were subject to the proviso: "if Jesus' claim that it is God who acts through him should prove to be true".

For the fourth act, Schwager then argued with ten arguments in as many subchapters that the historical-critical method had no way of explaining the claim of Jesus' resurrection other than through the divine resurrection action attested by the Gospels (Schwager [1989] 1999, pp. 119–35). Because God's raising of Jesus also confirmed his claim that God had acted in him against his opponents who accused him of blasphemy, Schwager's hypothetical assumption that God himself had actually acted in Jesus' actions could be verified.

Schwager thus confronted a very restrictive tendency of the historical-critical method in the German-speaking world in the 1970s and 1980s with the truth claim that God did indeed act in Jesus and that, consequently, God's action must be accepted as determining history. Schwager demands that theologians should apply the historical method "really [...] critically" in that "scholars are prepared, from the viewpoint of the 'object' which they come across, to put in question unproved assumptions in their own method" (ibid., pp. 119–20).

### 3. The Theological Method Underlying Schwager's Dramatic Theology

*3.1. The Self-Application of Jesus' Universal Claim of God's Final Action to Theological Science*

In terms of a theory of science, Schwager thus insists on the *self-application* of research content that is recognised as true to research practice with its respective methods. This demand is conclusive because the claim made by Jesus and testified to as such in the Gospels, that God acts definitively through him, is universal in such a way that, if verified, concerns all people comprehensively—including biblical theologians in their scientific work. When they encounter this claim of Jesus in their work and with their methods and as researchers recognise it as true—beyond any limited methods they may use—they find themselves just like the addressees of Jesus' proclamation at that time (and later the apostles, the Gospel, the Holy Scriptures and the Church) placed under this very claim. Like them, they find themselves in a situation of decision, a kairos (Schwager [1989] 1999, pp. 58, 109; Sandler 2014), which challenges them to accept or reject this claim that is acknowledged to be true: "The time is fulfilled, and the kingdom of God has come near; be converted (metanoeîte), and believe in the good news" (Mk 1:15, my translation). Such a conversion can therefore also require a "conversion of thought", as Paul puts it in the Letter to the Romans:

> "Do not be conformed to this world, but be transformed by the renewing of your minds, so that you may discern what is the will of God—what is good and acceptable and perfect." (Rom 12:2)

For biblical theologians who find themselves exposed to Jesus' claim that God ultimately acts in him, and who recognise that they themselves are subject to this claim with what they are currently researching, this results in an appeal for a "conversion of thinking". They find themselves challenged to transcend historical methods which, in their categorical exclusion of the possibility of God's action, are in a sense "conformed to this world" (Rom 12:2). This means that they can continue to use them as partial auxiliary methods, but at the same time they are challenged to open up to a more comprehensive perspective. Such a more comprehensive perspective of scriptural interpretation was presented by

Schwager in *Jesus in the Drama of Salvation* with his "a historico-dramatic interpretation of Jesus' destiny" (Schwager [1989] 1999, p. 17, cf. pp. 15–17).

### 3.2. Schwager's Reception of Anselm of Canterbury's Method of a "Conversion of Thought" for Jesus in the Drama of Salvation

Schwager not only demanded such a self-application of Jesus' claim (of a definitive action of God mediated by his work) to the way of doing theology from historical-critical exegesis, but also made it his own and carried it out in his theology. He did this by following the theological method of Anselm of Canterbury as elaborated by the French Anselm translator and commentator *Michel Corbin*.[6]

Corbin conceived of Anselm's theological thought as a dynamic movement of ongoing conversion (de Cantorbéry and Corbin 1988, pp. 36–42), which takes place in a "faith that seeks understanding" ("fides quaerens intellectum") by seriously engaging with the objections of unbelievers. Thus, the praise of God in his recognised greatness can only be sustained in the face of irritating objections "by crossing the opposite side of supplication in order to grow and expand in the crucible of trial" (ibid., p. 36, my translation). Corbin describes this as an existentially accomplished *dialectic*—not Hegelian but Aristotelian (ibid., p. 35)—which in this way corresponds to the *dramatics* of Schwager's theology, who worked his way through the seemingly irreconcilable contradictions of biblical statements while at the same time placing himself existentially under the claim of God's definitive action in Jesus Christ. In this sense, Schwager attributes to Anselm a "method of despair" (Schwager [1989] 1999, p. 51). This results in a continuous "conversion of thinking" (Schwager 1990, p. 201) in the sense of a partial rejection and correction of previous concepts and ideas in order to come closer and closer to the believed God as the one beyond whom nothing greater can be thought ("id quo maius cogitari nequit").

Anselm implemented this movement in several of his works, not simply in terms of individual predications of God—as with Thomas Aquinas, according to a "triplex via" of affirmation, negation, and surpassing—but by continually adjusting different, seemingly incompatible attributes of God. He did this above all with God's mercy and justice. In *Cur Deus Homo,* Anselm succeeded "in reaching the true biblical message by confronting the biblical passages about justice—through the mouth of his conversation partner Boso—with the other series of passages that deal with God's incomprehensible goodness, and this in a way that carefully avoids any premature compromise between the two series. He relentlessly deepens the discussion of justice until it is able to embrace the utterances about goodness in their entirety" (Schwager [1989] 1999, p. 6). In doing so, Anselm left behind initial contemporary notions of divine honour, to which modern critics of his doctrine of satisfaction relentlessly held him, in order to move towards an understanding of God's justice, which turns out to be "id quo maius cogitari nequit", in such a way that at the same time, and precisely because of this, God proves to be maximally merciful (Anselm of Canterbury 1988, pp. 468–70 = Cur deus homo II, 20).

In *Cur Deus Homo,* Anselm did not derive the necessity of salvation through the incarnation and crucifixion of Jesus Christ directly from biblical-Christological statements, but took the path of indirect proof by demonstrating to his dialogue partner Boso that a liberation of man from the burden of his sin "remoto Christi", that is, in disregard of biblical Christology, was unthinkable. In a "method of despair" (Schwager [1989] 1999, p. 51) Anselm piled objection upon objection, proving to Boso the inadequacy of each of his proposals for man's self-liberation without the redemptive work of Christ. In this way, Anselm finally demonstrated faith in the incarnate God-man who gave himself up guiltlessly on the cross as the only possibility of salvation, in a far clearer way than through figurative ideas, which, according to Anselm, were like "painting on clouds" (Anselm of Canterbury 1988, pp. 308–11 = Cur deus homo I, 4).

Schwager used an analogous method in *Jesus in the Drama of Salvation* to clarify the controversial question of God's action in history. Although—unlike Anselm—he used biblical texts throughout, he worked on an argumentative level similar to Anselm's "remoto

Christi", in that he did not derive the truth of Jesus' claim of God's final action through him from the revelatory texts, but only considered it in the manner of a hypothesis. Like Anselm, Schwager then demonstrated indirectly that the biblical texts and the historically verifiable events triggered by the Christ event cannot be adequately understood without acknowledging the truth of this hypothesis:

> "My method may appear initially to be contrary to Anselm's; but it is finally similar. From the beginning we shall take Jesus' claim as hypothetically true. From this presupposition we will ask whether and what sort of coherence is manifested in what Jesus spoke of as the action of God. Only in connection with the Easter reports will the question about truth be put directly rather than hypothetically." (Schwager [1989] 1999, p. 51)

From there, the methodological significance of Schwager's effort to achieve coherence between seemingly contradictory groups of texts becomes clearer. It corresponds to Anselm's aporetic effort to reconcile opposing attributes of God, especially his justice and mercy. For Anselm, an important criterion for approaching an understanding of God as "id quo maius cogitar nequit" with regard to his justice was that this justice should at the same time prove to be merciful beyond all (previous) thought. Similarly, Schwager sought a coherent interpretation of the groups of texts belonging to the five acts in such a way that God's action is shown to be merciful and just throughout, with his justice (not unlike Anselm's) being shown primarily in God's doing justice to humans in his freedom and responsibility. God does this in a way that transcends conventional notions of justice, in that his sovereign act of salvation restores the freedom of human beings entangled in sin—even at the risk that they will use this newly given freedom to turn against God's liberating offer of salvation and so harm themselves, others and creation (second act). Through the self-offering of Jesus Christ on the cross (third act), God absorbs this self-judgment of human beings in such a way that it does not drive them into the eternal damnation of hell. Rather, through Jesus' resurrection and the outpouring of the Holy Spirit (fourth and fifth acts), he opens up a new possibility of salvation.

## 4. Obstacles to the Reception of *Jesus in the Drama of Salvation*—And How to Overcome Them

Despite, or perhaps because of, Schwager's significant deepening of his method (compared to *Must There be Scapegoats?*), the scholarly theological reception of *Jesus in the Drama of Salvation* beyond Innsbruck Dramatic Theology (Wandinger 2016) has remained rather limited (Niewiadomski [1989] 2015, pp. 30–32). There are several reasons for this, two of which I would like to discuss in more detail.[7]

### 4.1. Dealing with Outdated Historical-Critical Positions?

One reason for the weaker reception compared to *Must There be Scapegoats?* was probably Schwager's extensive debate with biblical theologians who used the historical-critical method rather restrictively. However, they could not be convinced, while other readers found Schwager's argumentation too detailed, especially since this sharp form of historical-critical exegesis remained limited to the German-speaking world, and even there it soon became outdated from a biblical-theological point of view.[8]

Schwager's contemporary debate, however, touched on the still virulent problem in the philosophy of history of a contradiction between the historical and the dogmatic method, which Troeltsch had sharpened and in which he accused the latter of supranaturalism. For Franz Gmainer-Pranzl, "there is no question that Schwager's consistent questioning of the claim of 'God's action in human history' touches a nerve in modern philosophy of history, which [...] excludes any notion of a 'special action of God' from the field of historical knowledge" (Gmainer-Pranzl 2014, p. 54).

Moreover, Schwager was pursuing a concern of continuing relevance beyond his engagement with historically critical theologians when, as shown in Section 3.1, he demanded that they apply beliefs with a universal validity claim to themselves and to the methods of

their theological work. In their scholarly work with the testimony of the Gospels, they are called to make a "decision that concerns the scientific method itself" (Schwager [1989] 2015, p. 227, my translation; cf. Schwager [1989] 1999, p. 119), and this is true not only for exegetes but for all theological disciplines. People and communities who recognise Jesus' claim of a new divine action through him as true put themselves in a situation of self-judgment if they nevertheless reject this claim or evade a decision. This also applies to people who recognise this claim of Jesus as true in the course of their theological work, and not only in the field of biblical theology.[9] This could lead to resistance to the existential call of God and a consequent weakening of faith among academic theologians, theological faculties and their spheres of influence. A serious self-application of faith-relevant knowledge to theological practice and methodology, on the other hand, can lead to an ongoing conversion of thought and existence at the same time, as Schwager has developed for his own dramatic method in the wake of Anselm of Canterbury and Michel Corbin.[10]

It turns out that Schwager's detailed engagement with the historical-critical method is far from outdated, but has an exemplary significance for theological scholarship as a whole—in a lasting and as yet largely untapped relevance. This is inscribed in his dramatic method and especially in his historical-dramatic exegesis, which Schwager developed in *Jesus in the Drama of Salvation* out of consistent listening to the claim of a new action of God in Jesus Christ. In the sense of such listening in self-application and ongoing transformation of theological understanding, dramatic theology is essentially a *spiritual theology*.[11]

### 4.2. Fixing the Five-Act Drama of Salvation in a Rigid Scheme?

Schwager's *Jesus in the Drama of Salvation* is best remembered for its five-act scheme. It has proved to be a fruitful biblical hermeneutical approach, beyond specialist theological discussion, for perceiving tensions between biblical texts and statements as challenges and for overcoming them.[12] However, there is also a danger that the five acts will be misunderstood or misused as a rigid scheme which, in the course of its presentation, simultaneously freezes the drama of salvation and misery between divine and human action. Demanding theological questions, for example, in the interpretation of difficult Gospel texts, can then be defused by neatly assigning questionable statements to the 'right' dramatic acts on a conceptual level, without being challenged by the dramatic dynamics to a real conversion of life, thought and habitual ideas. By contrast, the previous presentation of the five-act structure should have made it clear that the real drama lies in the dynamic and catastrophic confrontations and transitions within, and not least between, the acts of the drama of salvation.[13] This drama is ultimately unresolvable because it reflects the irreconcilable mystery of God and the mystery of salvation. Detached from this, what remains is a rigid scheme which is almost inevitably undramatic, e.g., misunderstood as a chronological scheme of events, and inevitably rejected by scientific exegesis in such a truncated form. As a spiritual theology with a continued self-application (cf. Section 3.1), the dramatic method gains its forward-opening, inconclusive dynamic, which protects it from becoming systematic in a rigid way.

### 5. Relectures of Schwager's Five Act Drama of Salvation Towards a "Dramatic Kairology" of Divine Action

If one interprets the dramatic theology founded by Schwager consistently from its basic methodological approach of a "continuous conversion of thought" (Section 3.2), inspired by Anselm of Canterbury according to the interpretation of Michel Corbin, then it becomes an unfinished and unfinishable project, which must constantly work anew on the tensions between opposing biblical text groups and their reflections in contemporary church and social situations by constantly undertaking new transformations of previous concepts.[14] This allows for relectures and updates in the understanding of the five acts that are significant for a theology of divine action. In the following, I will reflect further on God's action of raising Jesus in the fourth act and its effects in the work of the Holy Spirit according to the fifth act, in order to illustrate from there how God's new action through

Jesus Christ in the first act draws people into minor or major kairoi of grace. Thus, they are initially transformed and, in this way, simultaneously challenged and empowered for sustainable self-transformation. I will show that this action of God is structurally similar to the action of God mediated by the Holy Spirit as described by Schwager in the fifth act—not only for the beginnings, but for the whole duration of the church up to our present. In this way the Gospel accounts of God's new action through Jesus Christ in the first act become typologically and criteriologically applicable to contemporary experiences of God's action. Thus, drawing on Raymund Schwager's dramatic five-act Christology, I develop a phenomenology of God's action through the events of an eschatological self-revelation of God that moves people a little way into the reality of a new creation (in the midst of the old) and thus opens for them a *kairos* that both challenges and enables them to live according to this new creation. In order not to lose the advantages of Schwager's dramatic approach in this further development of his dramatic Christology into a more phenomenological *dramatic kairology of God's action,* it was first necessary to go back to Schwager's methodological approach based on Anselm of Canterbury, with which he defined dramatic theology as an infinite process of conversion of thought (cf. Section 3). In what follows, I will move methodically along this path, specifically by mirroring the various dramatic acts within each other.

*5.1. Resurrection and New Creation: The Radical Newness of the Fourth Act in the Drama of Salvation*

The resurrection of Jesus from the dead means neither exclusively worldly—his mere resurrection in the sense of a return to this world (as a mortal again, as, for example, in the raising of the dead witnessed in the Gospels)—nor exclusively transcendent—Jesus' rapture into a heavenly "back-world" without effects on this world (Lohfink 2017, pp. 137–42). Only in a combination of the two does it unfold its biblically attested redemptive potential.

This twofold clarification of the meaning of the resurrection of Jesus Christ seems so self-evident in the first point—the rejection of his mere resuscitation—that Schwager does not specifically address it. The second point, that the resurrection of Jesus does not mean a mere heavenly rapture, but is a fact that affects this world and its understanding of the world, Schwager has fully dealt with in his rejection of a mere visionary hypothesis of the resurrection and in his detailed plea for the historical credibility of the biblical Easter account (Schwager [1989] 1999, pp. 119–24). For Schwager, the resurrection of Jesus is a radically innovative act of God that transcends this world and yet works within it to transform it. Only through a divine act of resurrection understood in this way could God at the same time confirm the claim, made by Jesus and disputed by his opponents, that his actions were consistently and from the beginning an expression of God's inner-worldly salvific activity.

However, the consequences of a belief in the resurrection, thus secured against the two opposing misunderstandings of mere super-worldliness or mere inner-worldliness, go much further. They affect not only the understanding of history, but the understanding of reality, the world and creation as a whole. The New Testament expresses this by not asserting Jesus' resurrection as a fact exclusive to him, but by calling the Risen One the "first fruits of those who have died" (1 Cor 15:20) and the "firstborn of all creation" (Col 1:15) (Lohfink 2017, pp. 142–47). The first means that in the resurrection of Jesus there is our confident expectation of the resurrection of all human beings after death. The second goes much further, in that it refers to the whole of creation and presupposes for it a radical new creation, which, moreover, does not only have an effect in a utopian imagined hereafter, but already in the present. For "if anyone is in Christ, there is a new creation" (2 Cor 5:17).

What is remarkable at this point is the following: This universalisation of belief in the resurrection to the whole of creation, which the Bible achieves, makes belief in the resurrection of Jesus comprehensible at all. For if we want to accept that Jesus Christ was raised from the dead, and not only in the sense of a resuscitation into this world doomed to death, but into the realm of a new creation (cf. Is 65:17) where death will be eliminated

forever (cf. Is 25:8), then we must ask ourselves how we are to conceive of creation as a whole so that this is possible at all. The confession that Christ has been raised from the dead can only be credibly defended if one does not strictly divide God's creative activity, by which he (1) initially brought the world into existence, (2) continually sustains and recreates it in a *creatio continua,* and (3) eschatologically recreates it, but understands God's act of raising from the dead as a radical re-creation, by which the new creation is founded and works into the old. According to the New Testament scholar Gerhard Lohfink:

> "God 'constantly' calls forth his creation out of nothing. And this creation out of nothing is owed to no one. It can only be understood as a continuous, unceasing act of God's loving kindness. This is precisely why it is theologically meaningful, indeed necessary, to call the resurrection of the dead a 'new creation', that is, to relate it to the creation of the world. For in this way it becomes clear that the resurrection of the dead is not an addition, which may or may not be, but rather, although it is pure grace, it belongs to God's plan of creation: from the beginning, creation is directed towards perfection, towards glory, towards a home in God." (Lohfink 2017, p. 143, my translation)

How such a perspective on the resurrection could develop in view of the Christ event can be understood from the apocalyptic context of the Jewish expectation of the kingdom of God and the belief in the resurrection at the time of Jesus (ibid., pp. 144–46). This was linked to a two-aeon scheme in which the dawning kingdom of God and a resurrection of the dead were assigned to the new aeon. Thereby, a radical break between the old and the new eon was taken for granted. John the Baptist shared this apocalyptic with his message of judgement that "the axe has already been laid to the root of the trees" (Mt 3:10) and combined it with the expectation of an imminent eschaton (Hahn 1998, p. 92).

Jesus' proclamation of the kingdom of God, on the other hand, contains a momentous shift when, in contrast to the Baptist, he proclaims that the kingdom of God has already arrived with certain events in this world of the old creation (Lk 10:20). Additionally, Jesus' announcements of his suffering, death and imminent resurrection on the third day must have caused confusion to the disciples, given the contemporary apocalyptic expectations of a resurrection (Mark 9:32; Luke 18:34). After all, it was hard to imagine that the world would end in the meantime.

At the latest with the disciples' confession that Jesus had risen, earlier apocalyptic ideas had to be transformed so that the new aeon or the new creation had already begun while the old creation was still going on. This "overlapping of aeons" (Sandler 2020; cf. Lohfink 2017, p. 145) was to be understood, moreover, in such a way that the new creation, beginning with the resurrection of Jesus did not take place in an afterlife unconnected with this world, but with impressive effects on this old creation condemned to sin and death. This was especially evident in the appearances of the Risen Lord, which completely reoriented the disciples, and subsequently the unfolding of Christianity and thus the world.

### 5.2. Schwager's "Second Reading of the Dramatic Doctrine of Salvation" in a Universal Perspective of Salvation History

In *Jesus in the Drama of Salvation* (Schwager [1989] 1999), Schwager did not yet take into account this new view of the whole of creation that resulted from the Christian belief in the resurrection. This changed when he attempted to understand original sin in the context of evolution and hominisation in his subsequent book *Banished from Eden: Original Sin and Evolutionary Theory in the Drama of Salvation* (Schwager [1997] 2006). In it, Schwager concluded that sin could evolutionarily "penetrate the natural human constitution" (ibid., p. 57).[15] This led him to a heightened problem of salvation, which now had to free humans from evolutionarily rooted, unsavoury natural dynamics.[16]

This led him, in a "second reading of the dramatic doctrine of redemption" (Schwager 1997b), to sketch anew the five acts of Jesus' drama of salvation from an expanded perspective. He now proceeded on the assumption that in an "extremely intensive compression of time" the entire history of humanity from its beginnings to its end was "comprised and

included" (Schwager [1997] 2006, pp. 58, 63–65). For the first act, Schwager now stated that Jesus "with the Basileia message [...] once again reached back to the beginning of humanity" (ibid.). This had happened in that Jesus revealed a God of boundless goodness and his original divine will for peace and reconciliation with reference to all humanity and reawakened corresponding buried hopes. In view of an evolutionarily rooted entanglement in holiness, however, the rejection of Jesus' offer of salvation in the second and third acts now takes on much greater weight: for the first act, Schwager now explained that "with his message of the kingdom of God Jesus referred again back to the beginning of humanity" (ibid.). This happened because Jesus revealed a God of infinite goodness and an original divine will for peace and reconciliation for all humanity, thus reawakening buried hopes:

> "But since Jesus appeared and proclaimed God's rule at a point in time when, over the course of evolution, sin had long lodged in the natural makeup of human beings, an acceptance of his message was no longer possible on a purely ethical level. Nevertheless, the kingdom-message remained a real possibility. To realize it a miraculous power was certainly needed, namely a faith that could move mountains (see Matthew 17:20; 21:21; Mark 11:22–23) and liberate and heal the human nature that was ill and imprisoned by evil (see Mark 1:21–2:12 and parallels)." (Schwager [1997] 2006, p. 60, my translation)

In the face of an evolutionary entanglement of guilt, however, the rejection of Jesus' offer of salvation in the second and third acts takes on much greater weight. From here, Schwager sought to understand Jesus' self-offering on the cross as a way "to heal the sickness of human nature" "through a mountain-moving faith that subverted the whole past history of sin" (ibid.). Schwager thus gained an argument that salvation is only possible through death:

> "If sin was deeply etched in human nature, then evil can only be overcome if this nature itself dies and is created anew by God. The death and resurrection of Christ thus turn out to be that radical divine response of redemption which, since the faith that moves mountains could not be awakened, was necessary in order to heal the long evolutionary history of sin from its very roots. Full salvation can therefore take place only through death, and an immanent or this-worldly presentiment of this salvation is only possible where people do not simply count on their good will and their own moral efforts. To the contrary, the old self must die with Christ (Romans 6:1–11; Galatians 5:24–5) so that the new self can be born in the power of the Holy Spirit (John 3:3–8). The process of redemption accordingly occurs as a dramatic recapitulation and victorious reversal of the long history of sin. The whole past of humanity is lived anew in the history of Israel and finally compressed in the drama of Jesus' destiny." (Schwager [1997] 2006, pp. 62–63)

Schwager thus drew on the strong biblical expression of dying with Christ (Rom 6:8) and being "born again" or "born from above" (cf. John 3:3), but how is it to be understood that "human nature itself dies and is created anew by God [...] in order to heal the long evolutionary history of sin from its very roots" (ibid., p. 63)? In his unpublished *Holland Manuscript* (Schwager 1997b) shortly after the completion of *Banished from Eden*, Schwager gave some further indications:

1.  On the first act, he wrote: "With the message of the incipient reign of God, Jesus reaches back, as it were, behind the history of sin";
2.  On the mountain-moving faith through which "the kingdom-message remained a real possibility" (Schwager [1997] 2006, p. 60), Schwager added in the *Holland Manuscript*: "*The biblical accounts of miracles* and especially the statement about the faith that can move mountains are therefore very important for a dramatic theology";
3.  Then, he said of the fourth act: "The new life does not come from an ethical conversion, the new life is a *new creation* that transforms the previous creation. The old self, the old man, the first Adam, the product of evolution, must die in order to be raised in and

with Christ to a new life and a new way of being. Here the dramatic theology meets completely the theology of Paul, who does not reckon with an ethical conversion, but speaks of the crucifixion of the old man and of the new creation." (ibid., my translation, emphasis mine).

The third point is entirely consistent with what I wrote in Section 5.1 about the consequences of belief in the resurrection for a theological understanding of reality and creation as a whole. There I was concerned with the question of how we have to understand reality and creation as a whole so that the belief—which Schwager has shown to be historically undeniable—that God raised Jesus Christ from the dead is also conceivable. The answer is the same as that given by various New Testament texts: namely, that Jesus Christ, as the Risen One, is the beginning of a new creation which will not come into being only after the fall of this world, but which already now has a profound effect on the 'old creation' which continues to exist. This happened in such a way that the disciples who remained behind after Jesus' crucifixion, as well as the emerging Christianity that followed, were completely reoriented by the appearances of the Risen Christ, in a way that the Pauline epistles express by saying that human beings already *are* a new creation in Christ (Gal 6:15; 2 Cor 5:17).

How we can imagine a new creation in Christ is shown by the outpouring of the Holy Spirit at Pentecost in Jerusalem (Acts 5; cf. chp. 2.2; 5.3). In addition, the miracles with which Jesus proved that the kingdom of God had already arrived (especially Lk 11:20), and to which Schwager attached great importance in the third reference from his Holland manuscript, can illustrate how God, through a new creation, makes new beginnings within the old creation. These can unleash a faith that moves mountains and thus overcome obstacles to salvation that reach into the evolutionary nature of human beings. To this end, it is necessary to understand at least the fifth and first acts anew and more deeply from an interpretation of the fourth act that is deepened from the perspective of a new creation.

*5.3. New Creation through the Holy Spirit: A Deeper Understanding of the Fifth Act Based on the Fourth Act*

What it means to be a new creation in Christ is demonstrated by the work of the Holy Spirit, released by the Risen One, who transforms the disciples from within so that they praise God and confess faith in Jesus Christ with a completely new courage and boldness. Schwager described this in *Jesus in the Drama of Salvation* for the fifth act (cf. Section 2.2), though without relating it to a new creation breaking in from the Risen One. In this way, the disciples were enabled to pass on the Holy Spirit to others in the sense of a renewed orientation towards God from within. Like Jesus in the past, they now proclaimed the incipient kingdom of God, but also—in the case of rejection of the divine offer of grace—an imminent judgement. In this way, they experienced an unprecedented growth of their proto-Christian community, but also resistance and persecution, even martyrdom. Thus, for Schwager, the fifth act stands for the whole period of the church, in which moments of the first three acts are repeated (Sandler 2012, pp. 126–32). We find this exemplarily in Acts 2–7, from Peter's sermon at Pentecost to the stoning of Stephen.

Yet, the first three acts are interwoven in a different way within the fifth act, as can be shown particularly in Peter's frank sermon at "Pentecost: Guided by the Holy Spirit", Peter managed to be both critical and supportive of the Jews present: he confronted them with the fact that they had rejected and crucified the Saviour sent by God (Acts 2:23; 2:36) and yet called them brothers (Acts 2:29). In a most astonishing way, he shook the listening Jews and at the same time extended his hand to them in solidarity:

"Now when they heard this, they were cut to the heart and said to Peter and to the other apostles, 'Brothers, what should we do?'" (Acts 2:37)

Now that they had accepted the suppressed truth of their past actions against Jesus, as revealed by the Holy Spirit through Peter, Peter could pass on to them Jesus' "redoubling of [his] readiness to forgive" (Schwager [1989] 1999, p. 136):

"Repent, and be baptized every one of you in the name of Jesus Christ so that your sins may be forgiven; and you will receive the gift of the Holy Spirit." (Acts 2:38)

Peter was guided in this by a spirit of critical solidarity in which justice and mercy grow with and upon one another (Sandler 2005, pp. 118–23): justice in faithfulness to God's word and truth, which led him to expose the repressed lies and violence in his listeners; and mercy in faithfulness to God's will for salvation, in a spirit of compassion for the broken situation of his listeners. Through an unwavering focus on the will of God—whose justice and mercy grow together in such a way that they identify God as "the One beyond whom nothing greater can be imagined" (Anselm of Canterbury, cf. Section 3.2)—Peter was able to convey to his listeners precisely this divine dynamic of a direct proportionality between justice and mercy. This is a behaviour that is barred to us humans because of our fallen nature, as Schwager explained in *Banished from Eden* (cf. Section 5.2).

Thus, without the support of Jesus (Lk 22:31), Peter failed miserably after the arrest of his Master. He fell into a fear that blocked any outspoken stand for God's truth in the face of his opponents, so that he could not help but deny his Lord. This fear gripped the disciples even after their encounter with the Risen One. Until the day of Pentecost, they hid and locked themselves in the upper room. It was only through a new work of the Holy Spirit that this wall of fear was literally swept away (Acts 2:2–6). Peter now had the courage to tell the unmanageable crowd of Jews present, who could turn into a violent mob at any moment, that they had rejected the Christ sent by God.

However, foolhardiness is also inherent in humans through a long evolutionary development. Mice have been observed to switch from a state of fear-induced shock to bold aggression, with which they attack the cat that has them in its clutches and escape from the irritated predator (Levine 2011, pp. 75–78, 88–89). Therefore, it would have been human, indeed all too human, for Peter to have launched into an angry speech against his opponents, thus offering the group of Jesus' disciples a new identity against the Jews. Fearful resignation in blind flight or freezing in fear and fanatical aggression are opposite possibilities open to human beings by evolutionary nature; however, the strong middle way of solidarity, which at the same time offers more critical truth, and of criticism, which at the same time offers more solidarity, so that both attitudes become one, goes beyond the evolved human possibilities. It is the breaking in of a divine reality (from the realm of the "new creation", completely 'filled with God') "beyond which nothing greater can be thought" and, in this sense, the *gift of the Holy Spirit*—"the Spirit of truth, whom the world cannot receive, because it neither sees him nor knows him" (John 14:17)—which the Acts of the Apostles reveal in an exemplary way with regard to Peter's Pentecost sermon.[17] The second chapter of Acts thus testifies to what Schwager's re-reading of *Jesus in the Drama of Salvation* was aiming at: how the Holy Spirit can touch people so deeply that their evolutionary deficits of original sin are to some extent overcome—at least for the limited time of a kairos event of grace (cf. Section 5.4).

In this way, the meaning of the Pauline phrase that "in Christ we are a new creation" (2 Cor 5:17) can be better understood in the sense indicated by Schwager: it consists of *a new being that is rudimentarily implanted in people, reorienting them from their inmost being towards God and at the same time towards the world,* so that they find God's glory in creation and in their fellow human beings, and praise God for it (Acts 2:11). So the evolutionarily anchored bonds of mimetic entanglement are loosened (Schwager [1997] 2006, pp. 57–72): The disciples, touched by the Holy Spirit, became free from fear of humans and from a judgmental "sideglance mentality" (Sandler 2021, pp. 50, 170, 175) and spontaneously sought the glory of God rather than the approval of others humans. Thus, they could uncover hidden midways between opposing, apparently mutually exclusive concerns, overcome polarisation and bring about a reconciliation that is sustainable in that it is not based on opposition to others. The Innsbruck research group Religion–Violence–Communication–World Order around Raymund Schwager formulated this later with the following "core hypothesis":

"A deep, true and lasting peace among people which is not based on sacrificing third persons and can exist without polarization onto enemies is very difficult or even exceeds human strength. If it nevertheless becomes reality, this is a clear sign that God Himself (the Holy Spirit) is acting in the people. This logic of incarnation is shown in the biblical message as well as numerous 'signs of the times' in human history." (Schwager and Niewiadomski 1996)

### 5.4. First Act: God's New Action through a Sovereign Setting of Kairos Events

However, this new creation is not a disposable possession. It is based on events of grace that have their *kairos* (Mk 1:15; Sandler 2014) and only bring about lasting change when people take advantage of this *kairos*. Without consciously intending it, and without being reflexively aware of it, people behave in such *kairoi* in a certain way according to the "laws of the kingdom of God",[18] which "the world does not know" (cf. John 14:17) and cannot fulfil. So, they receive God's grace in the form of a "doing experience".[19] In this way, they themselves have become in a sense *real-symbols* of the kingdom of God yet to come, or, respectively, of the new creation in the way of its rudimentary realisation. Moreover, they are this in the way of an *effective* sign: the new being in them is a ferment—in biblical metaphor: grain (Mk 4:26–31) or mustard seed (Mt 13:31)—for a lasting transformation of their fellow human beings and of the world, but above all, of themselves. God himself can act through them and thus become an event for others.

In this sense, Jesus could describe the people present in the Sermon on the Mount as the light of the world (Mt 5:14): as those who, in the light of the experience of the kingdom of God and of Jesus' powerful deeds and words, had to some extent become light themselves. Now, everything depends on them not to hide this light under a bushel, but to place it on a lampstand (Mt 5:15), by setting this rudimentary experience of being at the centre of their lives, so that from there, first their whole being and then the world can be illuminated and constantly transformed: "In the same way, let your light shine before others, so that they may see your good works and give glory to *your Father* in heaven" (Mt 5:16).

So, from the new creation founded in the Risen One and to some extent realised in the Holy Spirit, it can become clearer what Schwager repeatedly refers to as God's *new* action in the first act (Section 2.2). At the same time, from the proclamation of the kingdom of God by Jesus Christ in word and deed it can become clearer what it means to be a new creation and how an influence from the new creation on the old takes place through God's action. Through extraordinary acts of grace, such as healings, deliverances and miracles, and not less through the mere charisma of his presence,[20] Jesus made shine forth in limited spheres of reality and in an exemplary and emblematic way, a creation radically renewed by the kingship of God. In this way, objective gifts of grace are integrated into personal processes of a double self-giving (Sandler 2020), in which God—mediated through Jesus Christ and in experiences of the Holy Spirit for the time of the church—gives himself and at the same time sets the addressees free in their own being. Thus, for a limited time, while an event of God's saving action resonates within them, they are *given* to themselves anew (sich selbst neu *gegeben).* Precisely for this reason, they are committed to themselves (sich selbst *aufgegeben)* by experiencing themselves as called and at the same time as empowered to advance, through their own efforts, the new orientation that God has initiated in them.

In the Gospels, Jesus expresses this difference between the temporary emergence of being a new creation and its appropriation for a lasting transformation with the distinction between "healing" and "salvation". For example, the paralytic gets up or the blind man can see again, which Jesus comments with the words: "Your faith has saved you" (Lk 18:42 par. Mk 10:52)—saved, not only healed. The experienced physical healing in the restoration of earthly fullness of life makes salvation or redemption ("sōtería")—that is, a much deeper self-transformation towards eternal life—tangible in the sense of a real-symbol. The same faith relates to both—healing and eschatological salvation—by trusting God with everything. Thus, experiences of physical healing or spiritual liberation can strengthen

faith in the God who liberates us in our whole existence and makes us "fit for heaven". This happens when people return to Christ with a healing experience and praise God, such as the Samaritan who was the only one of ten people healed of leprosy to return. Jesus told him that his faith had saved him—over and above the healing that had already taken place (Lk 17:15–9).

This confirms Schwager's statement that despite a sin that "over the course of evolution, had long lodged in the natural makeup of human beings, [. . .] the kingdom-message remained a real possibility", which, however, could only be realised with "a faith that could move mountains [. . .] and liberate and heal the human nature that was ill and imprisoned by evil" (Schwager [1997] 2006, p. 60). It also becomes understandable that for this "the biblical accounts of miracles [. . .] are [. . .] very important" (Schwager 1997b; Section 5.2).

Thus, this offer of God's self in Jesus Christ, concretized in a particular salvific event (e.g., healing or deliverance), brings about a kairos (Sandler 2014)—in the sense of an opportunity for salvation that simultaneously enables and demands a deeper acceptance of God, within a limited time in which a salvific event still resonates powerfully enough to facilitate or even enable its active acceptance.[21] This explains the urgency that Jesus repeatedly exhorted in the Gospels (Lk 12:36) to make use of the respective kairos, combined with the necessary vigilance not to overlook a kairos and to recognise it immediately.[22] This urgency results from the fact that God's gift of kairos not only heals people outwardly, but also transforms them inwardly: people find themselves in a state of spontaneous agreement,[23] from which it is easier to accept the gift of grace and to carry out the necessary reorientation ("conversion"). With the fading of a kairos of grace, this help for acceptance also disappears, so that people may fall into fear and bury the gift offered (Mt 25:25).

For the first act of Jesus' proclamation of the kingdom of God, it emerges that God's offer of salvation is wholly good in itself—it is "a piece of heaven", a "mustard seed of the kingdom of God" or "of the new creation in the midst of the old", realised and released through God's new creative action. With a kairos of grace, God symbolically sets a *new beginning* in a limited realm of reality, in which he reaches back to the unbroken beginning of creation (Schwager 1997b; Section 5.2) and, at the same time, opens up an anticipation of the comprehensive new creation that is yet to come—in the manner of a *fulfilled* new beginning.

Thus, in the midst of the old, the new creation shines forth in the concrete events of the encounter with Jesus, not only as a promise of something yet to come, but is first experienced in limited events of salvation as a truly symbolic reality already present. As such, it is truly a gift, even if only temporarily.[24] In a kairological, exemplary, new-creative action, God, in a sovereign way that is completely unavailable to us—when, where, how and for whom he wills—measures anticipatory experiences of fulfilment with the character of promise, which are strong enough to tear human beings themselves out of the dynamics of catastrophe that are deeply rooted in nature, and to lead them in a new direction. Yet, this happens so fleetingly that these experiences of the new are not overwhelming, but can be accepted or rejected in a released freedom for radical self-transformation. God's redemptive action through the germination of the new creation in the old therefore proves to be 'minimally invasive' and maximally cautious.

## 6. Conclusion and Outlook: Divine Action in the Wake of Raymund Schwager's Dramatic Theology

*6.1. A Personal-Dialogical Understanding of Divine Action That Does Not Break with Classical Theological Attributes of God*

In this study I have sketched out a new approach to a theology of God's action by starting from Raymund Schwager's biblical doctrine of redemption *Jesus in the Drama of Salvation* and reconstructing in particular his dramatic five-act model of the Christ event as a theology of God's action. Such an approach is not completely new (Wandinger 2014; Gmainer-Pranzl 2014), since Schwager's dramatic theology—and in its wake the "Innsbruck Dramatic Theology" (Wandinger 2016)—is already, in its approach and concept, a theological theory of action which, in contrast to other theologies of action,[25] emphasises the

complex interactions between different agents, individually as well as collectively, and thus gives God's action a central place in complex human coexistence.[26] Schwager understood divine action as a personal, dialogical-interactive action, which is not exclusively limited to human beings, but in every case—even in natural wonders, which Schwager took seriously as signs of the new creation[27]—is directed to persons and thus always at the same time a revelatory action. In this respect, Schwager understood God's action in the same way as Karl Rahner, who, as a predecessor of Schwager, held a chair of dogmatics in Innsbruck until 1964:

> "God's action in the course of salvation history is not, so to speak, a monologue that God conducts for himself alone, but a long, dramatic dialogue between God and his creature, in which God gives man the possibility of a genuine response to his word and thus actually makes his own further word dependent on how man's free response has turned out." (Rahner 1997, p. 373)

For both Rahner and Schwager, this dependence of God on the response of human beings did not exclude the possibility that God might act in an absolutely sovereign way by making a new beginning with human beings and not only subsequently absorbing the rejections of his salvation initiatives, but initially taking them into account in a comprehensive plan of salvation.[28] For Schwager there was no doubt that God's action "always also depends on human reactions, although it is ultimately in no way determined by them" (Schwager 1997a, pp. 7–8, my translation).

In contrast to deism and to a semideism, which limits God's historical agency for reasons of theodicy,[29] Schwager had great confidence in God's ability to act. In this he also differed from pantheistic and panentheistic approaches, which attribute to God an omnipotence, but no intentional, distinguishable special action.[30] Although Schwager highly valued God's historical agency, he did not get caught up in fundamentalist and anthropomorphic notions of divine intervention (Gmainer-Pranzl 2014, p. 44). Through his action, God breaks neither the autonomy of inner-worldly realities according to natural law (Schwager [1997] 2006, p. 128) nor human will. Rather, he "incarnates" his action deeply into creaturely nature and human freedoms, so that it takes on different historical forms through different human reactions to it, which can obscure or distort God's genuine action. Thus, the "negative reactions to his [Jesus'] message […] brought about a completely new modality of its realisation, but did not cause it to fail" (Schwager 1997a, p. 8, my translation).

The fact that God can still realise his genuine, wholly salvific intention through the countless misunderstandings, rejections, appropriations and perversions of his action depends on different levels of his plan of salvation. Dramatic theology in and after Schwager is confident that God takes into account possible rejections and perversions of his action in advance in a more comprehensive plan (cf. Rom 11:25 and on this: Sandler 2023). By the same token, it follows from the methodological approach of dramatic theology that it is not possible for human beings—theologians not excepted—to decipher God's plans of providence and predestination in view of the manifold refractions of God's action in history. Schwager concludes from his concept of drama "that no theology can be designed from God's point of view—be it as incarnation theology or predestination theology" (Schwager 1997b). Dramatic theology is thus on the one hand opposed to the classical doctrines of predestination, as in Augustine and Calvin (Schwager [1989] 1999, pp. 4–8), but on the other hand it also differs from an "open theism", which is opposed to it. Dramatic theology contrasts with the advocates of open theism where they are convinced that they have to abandon the classical eternalistic idea of a supra-temporal providence of God in order to conceive of a God who acts in dialogue and responds to human beings, and that this is the only way to avoid the difficulties of the classical doctrines of predestination.[31]

## 6.2. Divine Action and Human Freedom

It is central to a dramatic-christological and kairological understanding of divine action to understand it essentially as liberating—so that it also implies a libertarian freedeom of

choice. In kairos events, God does indeed bring himself into the horizon of human freedom as a truly fulfilling goal—as in Augustine's understanding of *libertas*. However, this does not happen, as in Augustine's case, with an irresistible force. Rather, God reveals himself in a deeper way, which at the same time opens up to human beings the abysmal possibility of rejecting him more decisively than ever before.[32]

For Schwager, a divine liberation of human freedom in events of grace is soteriologically indispensable because human beings, in the sense of René Girard's mimetic anthropology, which Schwager shares, "are not autonomous. Either they are enslaved by passions or they let themselves be empowered by God to become master of their own house" (Schwager [1978] 1987, p. 171). A few years later, Schwager presented this position, which was basically critical of liberal theology, in a more differentiated way:

> The biblical scriptures speak clearly of freedom. But they also reveal a quasi-mechanical or compulsive character in human action. However, they do not attribute this kind of mechanics to a biological peculiarity of man, but interpret it—to speak with Ricœur—through a "symbolism of evil", through the enslavement of man's innermost spiritual freedom. (Schwager 1985, p. 373, my translation)

According to this, it is the fruit of events of grace, such as Jesus brought about through his powerful proclamation, and such as people experience again and again more or less explicitly as the working of the Holy Spirit, that God releases this creaturely basic freedom, which is directed towards himself, his fellow human beings, the world and himself, and claims it in a resulting decision situation, so that humans can and must decide whether to accept or reject this divine offer of grace.

By understanding God's salvific action as liberating in this way, Schwager gains a decisive advantage for a theology of divine action: he does not have to laboriously prove belatedly how a discernible action of God in history is compatible with a presupposed unbroken human freedom.

*6.3. A Kairological Approach to God's Action in the World and in Salvation History*

Starting from the first act of Jesus' proclamation of the kingdom of God, Schwager understands God's action throughout as a "*new* working" through Jesus Christ. This working proves to be new insofar as Jesus opens up a *new beginning* with his proclamation of the incipient kingdom of God as a pure reality of grace ("the year of the God's favour" (Lk 4:19) and its symbolic realisation in fully empowered acts of healing and liberation for people entangled in sins.[33]

The specificity of this new beginning is revealed more deeply by the Risen One, insofar as he is understood as "the *beginning,* the first-born from the dead" (Col 1:18) and, in this sense, as the origin of the new creation (cf. Section 4.1). From a deeper understanding of the fifth act and, within it, of Peter's Pentecost sermon, this new beginning has become clear as a Spirit-worked new human behaviour that transcends the hopelessly wrong alternatives within the limited horizon of an evolutionarily developed and thus culpably impaired human nature (Section 5.3). Proceeding from the fifth act, we have been able to understand God's new action in the *first act* of Jesus' proclamation of the kingdom of God as a sovereign divine orchestration of events of grace, by which God opens up to human beings a part of the new creation and thus initially transfers them into a new being (cf. Mt 5:14). In so doing, God opens up the possibility for them and calls them to actively accept this initial "being of a new creation" for their lives, in order to prepare themselves for an eschatological, full entry into the new creation. This ever-new saving action of God, who sovereignly places people in smaller or larger kairos experiences, is ideally witnessed in the first act of Jesus' proclamation of the gospels and deepened in the fifth act—the time of the church—through the work of the Holy Spirit. From this point of view, we can understand God's saving action not only for Christians but for all people in smaller and larger kairos experiences.

In a theology of grace, these events, which God in absolute sovereignty distributes like a sower among and into people when, where and how he wills, can be interpreted as

God empowering people and communities to participate in their own salvation.[34] Such events of grace can break into people's lives in earth-shattering ways, whether individually (as in Paul's Damascus experience: Acts 9; 22; 26) or in community (as at the Feast of Pentecost in Jerusalem: Acts 2). However, they can also happen in such a subtle way that they are immediately forgotten or even passed by carelessly. The possible places and times where they may invade are so varied that they encompass the entire history of creation and salvation. Everything that exists is created by God in such a way that he can make it a medium, a real-symbol or a "sacrament" of the breaking in or shining through of the new creation—and in it of himself—in the midst of the old.

### 6.4. God's Action as Kairos Event, New Creation and Resurrection: Three Theses

In conclusion, on the basis of and in the wake of Schwager's dramatic Christology, I will express this approach to a theology of divine action in three theses:

(1) God's action in the world and in the history of salvation consists essentially in a *kairological salvific action* in which God gives people a "piece of heaven" by opening up in a small part of their world a piece of its heavenly form of perfection[35];

(2) This kairologic salvific action of God is to be understood as a *new creative act* by which God brings humans a step closer to a *new beginning*, thus giving them the possibility of continuing or rejecting this transformation of their own accord;

(3) As shown in the context of the various acts of the drama of salvation, this new creative action, as we find it in the first and fifth acts in Jesus' empowered proclamation of the kingdom of God and in the experiences of the Holy Spirit, is *grounded in God's creative act of raising Jesus Christ from the dead.* "In him", through the Holy Spirit, we are given access to the new creation in the midst of the unredeemed old. This does not always happen, but it does happen time and again. In this way, we are rudimentarily transformed into a new being, so that we *are* initially a new creation (2 Cor 5:17; Gal 6:15), enabled and challenged to live accordingly as new people.

### 6.5. Outlook

In this essay I have started from the present difficulties of a Christian spirituality that gives much credit to God's saving action concretely for one's own life and for the world. In order to overcome the corresponding polarisation between a trust in God that has become weak by biblical standards in mainstream churches, and a stronger but often exaggerated confidence in God in charismatic and evangelical renewal movements, I have called for a theology of God's action that fulfils four criteria: (1) a biblical-theological foundation with a corresponding biblical hermeneutics, (2) theological coherence and competence in dealing with relevant problems, (3) relevance to life and vividness without simplification, and (4) spiritual fruitfulness.

The readings and rereadings of Raymund Schwager's dramatic theology undertaken here were intended to develop such a theology. In a subsequent essay, this concept will be tested against the four criteria mentioned. This will include, in accordance with the second criterion, situating the approach presented here in relation to other theologies of God's action and the theological questions they raise and to which they respond. The phenomenological-kairological development of Schwager's dramatic theology, elaborated here primarily in discussion with biblical texts from the Gospels, will have to be further clarified and deepened in dialogue with philosophical and theological phenomenologies of the event (cf. Section 5.4; note 21). In addition, it will be necessary to examine the guiding metaphors obtained here in terms of criteria 3 and 4 and to clarify them further theologically: this concerns in particular the guiding idea of a God who acts for our salvation by "sowing the seeds of the kingdom of God in our hearts". On the basis of the approach presented, necessary theological distinctions will also have to be made on controversial and polarising issues, not least on the question of a God who works miracles (Blay 2022; Sandler 2022). And it will be necessary to broaden, at least to some extent, the biblical-theological basis,

which here has remained almost exclusively limited to the Gospels and the first chapters of Acts.

Such a theology of God's action, theologically broader in scope and biblically developed from the ground up, will have to prove itself against the criteria of relevance to life and clarity without simplification (3) as well as spiritual fruitfulness (4) in the face of the questions I raised at the beginning of this essay: namely, in the recovery of a biblically based faith in God's saving power, which, in the face of concrete personal, social and global challenges, trusts God unconditionally with everything, but without fixing him on longed-for interventions and without holding out the prospect of such divine interventions to others if they only pray correctly and decisively enough, as is often the case in neo-charismatic movements.

Jesus found such faith, to his amazement, in a centurion (Mt 8:10); he repeatedly missed it in his disciples, whom he accused of having little faith (oligópistos) (Mt 8:26; 14:31; 16:8; 17:19; Lk 24:25), and he radically demonstrated it himself in his prayer on the Mount of Olives (Mk 14:36). The golden mean of an unrestricted faith that does not want to bind God in any way is described in a particularly impressive way in an Old Testament story in the Book of Daniel. There, Shadrach, Meshach and Abed-Nego explain to the great king of Babylon, who threatens to throw them into the fiery furnace if they do not worship the pagan statue:

> "O Nebuchadnezzar, we do not need to give you an answer concerning this matter. If it be so, our God whom we serve is able to deliver us from the furnace of blazing fire; and He will deliver us out of your hand, O king. But even if He does not, let it be known to you, O king, that we are not going to serve your gods or worship the golden image that you have set up.". (Dan 3,16–8, New American Standard Bible)

**Funding:** This research received no external funding.

**Conflicts of Interest:** The author declares no conflict of interest.

## Notes

1  Cf. (Koziel 2022). Magnus Striet (Striet 2021) addressed this problem polemically in his essay "Theologie im Zeichen der Corona-Pandemie". He was keen to criticise the typical responses of a "bad theology" that sought to blame people for evil in general and COVID-19 in particular, and he found few public statements that linked God and COVID-19 at all. An exception is the statement, emphasised in mainstream and also free churches, that corona is not a punishment from God (Koziel 2022). This is certainly highly desirable, but it leaves the question unanswered: What can be said positively about God in the face of Corona? What can we trust him to do in the face of these humanitarian disasters?
For a further perspective that takes a comparative look at the Anglo-American and European regions, I refer to the succinct statement by the Australian-British sociologist Bryan S. Turner (Turner 2021), who, against the background of earlier theodicy questions triggered by catastrophes, considers the religious theodicy question to be closed: "Religious theodicies rarely or never prove convincing in a secular age, and therefore people in search of meaning may seek out secular, specifically political, explanations of injustice, suffering, and disaster" (Turner 2021).

2  Otto Kaiser already emphasised this in the subtitle of his comprehensive *Theology of the Old Testament*: "The Path of God in the Old Testament from the Lord of His People to the Lord of the Whole World" (Kaiser 2013, my translation).

3  Unless otherwise indicated, Bible texts are quoted according to the New Revised Standard Version (NRS). All emphases in Bible texts are mine.

4  See the song "Mighty to Save" by the charismatic evangelical band Hillsong Worship with 15 million hits on youtube: (Hillsong 2017).

5  According to Schwager, this applies only for very few, obviously archaic texts (Schwager [1978] 1987).

6  See especially (de Cantorbéry and Corbin 1988) and on this Schwager's review (Schwager 1990). Michel Corbin's important interpretation of Anselm has not yet been translated into English or German and has therefore received little attention. For the German-speaking world, apart from Schwager, it is mainly Martin Kirschner (Kirschner 2013) who has taken Corbin into account.

7  A third obstacle to a wider reception of Schwager's theology is his reception of René Girard. Girard's theory of mimesis and scapegoating was increasingly perceived and rejected as a monomaniacal total explanatory model, and because of his strong reference to Girard, this prejudice hit Schwager in one blow (Moosbrugger and Peter 2016, p. 36). On the one hand, Girard himself has been misunderstood because he never connected his far-reaching verification attempts with a philosophical claim

to universal validity, so that they can also be understood as a test of the scope and limits of a hypothesis. Secondly, Girard's theory was consistently received by Schwager and in his wake as a theology of original sin (Miggelbrink 2004, p. 180). This makes it possible to assume a universal relevance of mimetic phenomena—in the sense of an "existential" (Rahner [1976] 1999, pp. 114–15)—without at the same time assuming that human beings are *entirely* determined by them. Third, it is ignored that Schwager did not simply adopt Girard, but strongly influenced him himself, especially with regard to his original "anti-sacrificial" interpretation of the New Testament, from which he explicitly distanced himself under Schwager's influence (Moosbrugger 2014).

8    This is how Markus Bockmuehl criticised the English translation of *Jesus in the Drama of Salvation*: "To what extent does his exegesis depend on the dated, mainly German scholarship of the 1960s and 1970s, untouched by the Third Quest, that makes up the bulk of his citations?" (Bockmuehl 2000, p. 134).

9    As far as I can see, Schwager has not worked out what such a self-judgment consists of on the level of a theological practice that eludes decision. Jesus' words of woe against the Pharisees and scribes (Mt 23), who are in a sense the forefathers of today's theologians, would have to be evaluated. The extent to which such a self-judgement is connected with a far-reaching existential irrelevance of theology and with a rapid decline in the number of students in theological faculties (which certainly has other causes as well) would have to be examined separately.

10    For a further elaboration of the significance of a conversion of thought for theology, one would also have to consider the connections between Schwager's dramatic theology and Bernard Lonergan's "method in theology" (Lonergan 1971, pp. 130, 237–45). Cf. Wandinger (2007).

11    In the wake of the historical-dramatic biblical hermeneutics developed by Schwager in *Jesus in the Drama of Salvation,* a hermeneutic circle between theoretical knowledge and practical self-application emerges, which drives the understanding of biblical revelation with its apparent contradictions (for example, between God's justice and mercy) towards ever new transformations.

12    Cf. the numerous dramatic-theological interpretations of the Bible, especially by Raymund Schwager, Józef Niewiadomski, Nikolaus Wandinger and Willibald Sandler online in the *Innsbruck Theological Reading Room* under the categories Sermons and Articles: https://www.uibk.ac.at/theol/leseraum (accessed on 8 March 2023).

13    Schwager attached great importance to this. He wrote to the editor Gerbert Brunner of the Herder publishing house about Jesus in the *Drama of Salvation*: "The purpose of this model is, on the one hand, to give the individual moments (from the message of Basileia to the sending of the Spirit) in the New Testament events their full weight and, at the same time, to include them in a comprehensive plot (with numerous actors)" (Letter of 26 February 1989, RSA II, 7), quoted after (Niewiadomski [1989] 2015).

14    New transformations of the understanding of judgement and the apocalyptic can be found in (Schwager 1997a).

15    Schwager took up Wolfhart Pannenberg's assumption that there is a "basic state of natural constitution of perverted life" in the human individual (Pannenberg 1991, p. 295; Schwager [1997] 2006, p. 55). With René Girard—in "a more detailed extension of the Girardian perspective" (Schwager [1997] 2006, p. 71)—Schwager already assumed for the evolutionary ancestors of humans a desire shaped by imitation, which in the course of hominisation became increasingly unbounded and could thus lead to deadly rivalry. At the same time, the nascent human being developed a spiritual capacity and an initial scope for freedom and responsibility, combined with a transcendence that made him open to a relationship with God in the reception of grace and revelation and in their grateful acceptance or in their rejection. Within this evolutionary process, the Fall must be understood as a successive, increasingly culpable lagging behind God-given possibilities to direct mimetic desire into salvific paths. In this way, humans became increasingly violent in the process of their hominisation and cultural development, with increasingly culpable deficits affecting the evolutionary factors of hominisation, such as brain growth, and thus becoming ingrained in human nature.

16    "Only by passing over into a new community, which encompasses all their desiring, is a true renewal possible" (Schwager [1997] 2006, p. 40).

17    In the course of the Acts of the Apostles it becomes clear that even this pneumatic fruit of Jesus' gift of redemption does not automatically and necessarily lead hardened people to conversion. Rather, the inwardly transforming Holy Spirit creates a new kairos that can also be rejected. Unlike Peter, Stephen did not reach his audience with his speech, but was stoned to death (Acts 7:58). The combination of second-act criticism and first-act solidarity in a strong form that allows both to grow together is a gift of the Holy Spirit that helped Peter to his evangelistic success. However, the same gift of the Spirit led Jesus to the cross and Stephen to martyrdom in the footsteps of Jesus.
Putting all this together, we see how the Holy Spirit, released by the risen Christ, completes Jesus' work of redemption by a threefold action, allowing the work of his self-offering—as subjective redemption—to reach the very innermost of human beings: firstly, the Holy Spirit confronts people with the repressed truth of their guilty history; secondly, he reveals to them that Christ has already forgiven their guilt through his gift of himself; and thirdly, he opens to them a path of repentance and a new beginning in following the crucified and risen Christ. In this never compelling but highly effective threefold way, God in the Holy Spirit brings about profound transformations in people and in the church as a whole (Sandler 2011, pp. 111–13).

18    These include Jesus' antitheses in the Sermon on the Mount. Cf. on this (Sandler 2023).

19    Cf. Karl Rahner's remarkably sober examples of "experiences of grace" (Rahner [1954] 2006, pp. 484–85) or "experiences of the Holy Spirit" (Rahner [1978] 2007, pp. 48–51), each with two astonishing features: (a) they are not experiences that take hold of us from outside, but practices that we find in ourselves, and (b) the examples thus marked as "doing experiences" (Sandler 2021, pp. 86, 127) are perceived less as exalting, but rather as painful and perhaps even with a certain shame.

20    The effect of Jesus' mere charisma is particularly emphasised in the narratives of the calling of the disciples (Mk 1:16–9). For the discussion of Pentecostal and neocharismatic faith in God's saving power, it will be important not to link it one-sidedly to extraordinary miracles. It should be noted that in the Gospels even the great miracles are interpreted as mere seeds of the kingdom of God.

21    A further theological elaboration of these connections, which I have planned for a later essay, will have to deal with the specific temporal structure of kairos and event with respect to divine action. Cf. Caputo (2006) with his reception of Derrida's phenomenology of the event for theology, and—without favouring the concept of the event—Levinas with his concept of the trace. (On the importance of Derrida and Levinas for a theologically relevant understanding of event, cf. Zeillinger (2017)). As a result, an event transcends our categories and can therefore only be marked as an interruption in our chronological time, but cannot be assigned a specific extension. I thank Reviewer 1 for his justified criticism in this respect. On the other hand, the biblical texts, especially the Gospels, as discussed in this chapter, show that God's action resonates in our time and history, opening up limited windows of opportunity within which we alone can accept or reject God's offers of grace. Accordingly, a kairos has a limited time, which can also be missed. In order to take both into account, I no longer identify kairos and event (with kairos as the time for an event and also as an event that has its special time, cf. Sandler 2014), but rather speak of a divine event of grace that *has the effect* of a kairos in our lifeworld and time.

22    Jesus repeatedly emphasises this vigilance in his sermons (Mt 24:42, 25:13, Mk 13:34.37, etc.) and the Gospels make it clear in Jesus' exemplary narratives of a happily perceived kairos—for example, in the Zacchaeus pericope with kairos markers such as "today" (Lk 19,5.9) or "quickly" (Lk 19,5.6)—as well as of an embraced or missed kairos (e. g. the parable of the talents, in which the first servant uses his talent immediately ("euthéōs": Mt 25:15), and then the second ("hōsaútōs": Mt 25:17), unlike the third, who hesitates too long (with three verbs of withdrawal: "he went away, [...] dug [...] and hid": Mt 25:18) and so became afraid.

23    This is especially true in the summative story of a missed kairos in Jesus' sermon in his hometown of Nazareth (Lk 4:16–30): "All testified for him" (Lk 4:22a, my translation), as the very first response (Sandler 2014, pp. 11–2). This spontaneous assent corresponds to the connaturality of authentic, God-given experiences of salvation, so that it occurs more quickly than resentment, which can also be almost reflexive. This initial assent then reinforces a possible later rejection of the offer of salvation in the direction of *self-judgment:* what is then rejected is not merely a divine offer of salvation brought in from outside, but this offer of salvation as something originally recognised as good and accordling accepted.

24    Jesus' healings may have been permanent, but they are not identical with, and only symbolic for, a salvation that concerns the whole of existence.

25    For example, in contrast to the theology of communicative action developed by Helmut Peukert and inspired by Jürgen Habermas (Arens 1992). Cf. Schwager's response to this in (Schwager 1992, pp. 356–57).

26    In this last point, Schwager's dramatic theology touches on the theodramatics of Hans Urs von Balthasar. In contrast to Balthasar, Schwager and the dramatic theology of Innsbruck emphasise the importance of human actors. This has consequences, especially for soteriology. Cf. Balthasar's criticism of Schwager (von Balthasar 1980, pp. 288–91), which is, however, only based on *Must There be Scapegoats?*, as *Jesus in the Drama of Salvation* had not yet been written.

27    Cf. (Schwager 2001), https://www.uibk.ac.at/theol/leseraum/texte/80.html#20 (accessed on 8 March 2023).

28    According to Nikolaus Wandinger, "it can be assumed of the divine author [...] that he accepts the opposing will [of people to the divine offer of salvation, W.S.], and at the same time that he knows this opposing will and incorporates it into the overall course of the play in such a way that its overall sense is maintained" (Wandinger 2014, p. 201, my translation). This corresponds to the five-act drama, according to which God overcomes the catastrophe of a divine offer of salvation rejected in the second act by new salvific initiatives in the fourth and fifth acts.

29    As described by (von Stosch 2006, p. 102; 2018, pp. 26–28).

30    Reinhold Bernhard attributed a tendentious deism and also pantheism to what he called the "sapiential-ordinative model of God's action" (Bernhardt 1999, p. 441), and he described the "model of an operative presence", which he advocated and developed, as panentheistic (ibid.).

31    With its emphasis on the absolute sovereignty of God, Schwager's theology of God's action would have to be classified as eternalist, and in this respect differs from most forms of open theism. However, Schwager never explicitly addressed open theism; he died in 2004. A mediating position, which also takes Schwager's dramatic theology into account, has been developed by Johannes Grössl (Grössl 2015).

32    This is made possible, on the one hand, by the fact that God does not reveal himself directly, but through the mediation of creation and in a symbolic reference to himself. On the other hand, it happens through a temporal structuring of kairos events into different phases: a possibly overwhelming experience of God, which breaks open a guiltily (by original sin) narrowed horizon of human knowledge and freedom towards God, is followed by phases of a relative self-concealment of God. It is in this second period of a fading impression of God that humans can and must make a choice. This is the moment when faith can and must prove itself by decision, according to the word of Jesus: "Blessed are those who do not see and yet believe" (Jn 20:29), *after* they have already seen. Cf. (Sandler 2021, pp. 300, 312; 2023, the final chapter).

33  The kairological approach to God's action presented here is thus close to Saskia Wendel's idea of thinking God's action with the help of Hannah Arendt's concept of "natality"; cf. (Wendel 2013). For the dramatic-kairological approach to a theology of God's action presented here, it might be fruitful to reflect further on this connection between kairos, event and nativity.

34  In this way, we approach from a different angle Karl Rahner's statement "that there is no true and real opposition between external salvation and a so-called self-salvation, when in a Christian understanding of God it is clear from the outset that the 'self' is a setting of God himself" (Rahner 2002, p. 347, my translation). This absolute sovereignty of God is shown in the Gospels by the fact that Jesus is not the master of where he can work miracles (Mk 6:5). It is the Holy Spirit who guides and impels him, and this Spirit does not usually speak to him directly, but through the needy people who astonish him with their faith, surprise him with a secret touch, or arouse a truly supernatural—and at the same time physical and social—compassion. On the specific Greek word used for this, "splanchnizomai", meaning "to be shaken in the bowels", cf. (Sandler 2021, pp. 147–48).

35  Of course, there are other forms of God's activity and action in the world that are not directly and exclusively directed at human beings. In a widely used typology, Kessler distinguished four basic forms or stages of God's action: *first,* "God's unmediated creative action" in the sense of a *creatio ex nihilo* (Kessler 2006, pp. 97–98; here and in the following my translation); *second,* a "creaturely mediated general and continuous creative action of God in the world" in the sense of a *creatio continua* (ibid., pp. 98–100); *thirdly,* a "special (innovative) activity of God in the world mediated by human agents", the full form of which he sees "in unique concentration and definiteness" God's action "in the figure and history of Jesus of Nazareth" (ibid, pp. 100–1); *fourthly,* a "radically innovative action of God in resurrecting the dead and completing the world, which is not mediated by human action" (ibid., pp. 102–3). The dramatic-kairological approach presented here corresponds to Kessler's view insofar as for him only the third stage with its Christocentric basic form reveals an action of God in its full form, which, according to Kessler's definition of action (ibid., p. 93, as well as Kessler 1995, p. 287 with G. H. von Wright) also includes a discernible intention of the subject of action. Kessler considers this criterion to be fulfilled for the first and second basic forms only in retrospect from the Christocentric third stage of God's action. For Kessler, the fourth basic form of God's action of resurrection and consummation in order to bring about a new creation is that which is "fundamentally inaccessible and unimaginable" (ibid., p. 102), which in turn can only be expressed and believed by us through faith in God mediated by Christ (third stage)—as the "sharpest expression of the living, effective reality of God in general and is its outermost proof" (ibid.). For Kessler, these basic forms represent "four irreducible categorical levels of God's activity" (ibid., p. 97). Without fundamentally denying this irreducibility, the dramatic-kairological approach presented here goes decidedly further than Kessler's Christocentric focus on the third basic figure of divine action in mediating, with a Christological focus, the four basic forms of God's action: first, by the fact that the claim of God's action through Christ (with Schwager in the first act of Jesus' salvation drama; with Kessler in the third stage of his typology) cannot be maintained at all without the recognition that Christ has already risen and thus the new creation has already begun as one affecting this world (cf. Section 2.3). Secondly, in Section 5.1 I argued that faith in the resurrection of Jesus can only be sustained if creation and salvation history are understood in a fundamentally broader perspective, so that we understand the new creation, the kingdom of God and eternal life as one common reality to which our world is open from within, so that this openness can again and again be actualised by God through events of his self-revelation in the way of special action—namely, in a symbolic way, so that Kessler's second basic form of a "creaturely mediated general and constant creative work of God in the world" finds an intentional and personal form of expression, which consequently elevates God's work to action in the full sense.

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
