# Peer review of "Divine Action and Dramatic Christology: A Rereading of Raymund Schwager’s Jesus in the Drama of Salvation"

_religions, doi:10.3390/rel14030390_

Round 1

Reviewer 1 Report

The article operates within the framework of the Innsbruck discourses on a dramatic theology, which it takes up in an up-to-date and well-informed way. As the title and abstract precisely state, it offers a re-reading of Raymund Schwager's "Jesus in the Drama of Salvation", which he also embeds in the discussions of his approach and Girard's reception, in order to profile a kairological model of divine action from here. The connection of Christology and pneumatology succeeds in a very plausible way through Schwager's 5-step model. Innovative and argumentatively strong is, on the one hand, the reference to Anselm's method, in which the dialectic of God's mercy and justice and the logic of the ever greater God lead (ever anew) into a conversion of thinking and take the event of revelation seriously in a structure-forming way (with important reference to Michel Corbin). On the other hand, I am particularly convinced by the kairological/event-based model of God's action in connection with the necessary subsequent free human response, which decides whether the kairos is grasped and transformation processes are initiated. In my opinion, some central systematic statements and formulations in this context should be reconsidered: As a moment, kairos should be thought of as an event and interruption of time and not as a short time interval: so it seems wrong to me to speak on p. 16 of "kairos as limited time"; and of a gift "even if only temporarily" given: Here reflections on the temporality of kairos, on the thinking of the event and on the logic of the gift could be included; at least the wording should be reconsidered. On p. 18 "God moves people a little way into" seems a reification / limitation of God's Gift, again conceiving the event as a short interval of time.
The important remark on page 15 "So they receive God's grace in the form of a "doing experience" could be linked to the approach of a "performative theology" as being proposed for example by M. Kirschner (also with reference to Anselm: Gott größer als gedacht 463-467, further developed in other essays) or by G.M. Hoff (Performative Theologie, Stuttgart 2022); the logic of the event and the human response is developped for example in: Zeillinger, Peter (2017): Offenbarung als Ereignis. Zeitgenössische Philosophie, die Rede von Gott und das Sprechen der Bibel. . In: SaThZ 21, pp. 25-101.).
It is not necessary to further develop those hints, but in my opinion it would further improve the impact of the article, to go beyond the discourses of dramatic theology relating them to other approaches.

on page 12/13 is a Duplication of a sentence ("and this happens..."); also on p. 16 ("although not exclusively limited...") - this should be corrected   

All in all, in my opinion it is an excellent, well elaborated article,the further references do not necessarily have to be incorporatedt sich i

Author Response

Dear reviewer,
Thank you for the encouraging feedback for my project. 
Your objections to the understanding of time as a kairos in relation to an event of God‘s grace were very helpful to me. I had saved this topic for the subsequent essay I announced in this article. When your expert opinion reached me, I was just in the process of familiarising myself with the literature you had mentioned to me. The volume of the Salzburg Theological Studies with theological explorations on the event and the extensive essay by Peter Zeillinger was just on my desk. I was also working on Caputo‘s theology of the event („The Weakness of God“) and Hanna Arendt‘s concept of natality in order to continue and deepen my approach presented here in a subsequent essay. Therefore, I can very well understand the problem and danger of reifying the concept of kairos and event that you mentioned. I have now refined my essay in this respect by distinguishing between event and kairos. Cf. especially footnote 21 in the corrected version.
For the following essay, I would also like to consider again the scientific-methodological approach of Martin Kirschner, who can convincingly mediate conceptual-theological, transcendental and phenomenological approaches with each other, precisely starting from an interpretation of Anselm of Canterbury with Michel Corbin.
Mediating these different approaches, which are often played off against each other, is a task that is important for a theology of God‘s action with its different orientations in analytical, transcendental (with Karl Rahner) and phenomenological theology. Martin Kirschner‘s habilitation thesis is a great help here.

Reviewer 2 Report

This article is well-written, offers genuine contributions to theological thought at the intersection of Schwager’s work, dramatic theology, and divine action, as well as some contributions in each.  Schwager’s theology and its development are explained astutely and articulately; those sections were a pleasure to read.  The answers to the two chosen objections are less clear and crisp, but seem sound.

The application of the method of resolving seemingly disparate concepts is thoughtful and significant for contemporary theology, as is the cairological development of Schwager’s dramatic theology.

The author knows and draws upon the significant writings of Schwager himself and the literature about his theology, particularly in German, though less so in English (and I don’t know what’s available in French or other languages).

On this basis, I would recommend the article for publication with minor revisions.

The improvement that I would label as essential is that the article needs to tie all of the parts together more coherently.  Specifically, the application of Schwager’s/Anselm’s method to engaging in “repeated new readings” does not seem to belong in an article that is centered on developing a cairological dramatic theology of divine action.  I think that the two can be woven together coherently within this work, but currently that is not made clear.  Even the abstract simply describes the two separately.  My suggestion is that the method of “repeated new readings” can become simply a means of advancing the argument rather than a key point that bears mentioning in the abstract and introduction.

In addition, here are a couple further considerations:

As the article progresses, the reader needs a couple reminders of the main point of the article (and having a clear, single thesis will help).  Currently, the thesis (or theses) are stated at the beginning and repeated at the end, but seem completely forgotten in the 15 pages in between.

The sections of the article in which the author develops Schwager’s thought (as opposed to simply explaining it) are a little overly verbose and repetitive.  The author’s points could be made more concisely and clearly.

Consider applying the same both-and method to the seemingly counter claims that Schwager makes that salvation is pure grace/“unconditional” yet also depends upon the rational creature’s response.  These must be purified, too, so that we come to understand grace better, as already strongly implied in the article.  When it comes to the relation of the world to God’s action towards the world, “gratuitous” and “deserved” need deepening in light of each other in the same kind of hermeneutical circle that Schwager finds in Anselm’s treatment of justice and mercy and his own treatment of the words of Jesus pre- and post-rejection.

Few references to theologies of divine action have been incorporated; that would be good to rectify.

Bottom of page 4: Schwager’s analysis of the history of soteriology in his collected articles in Der wunderbare Tausch suggests that overall he would not call “traditional” the idea that the Father acts on the Son on the cross (punishes the Son in our place).  There is a line in the theological tradition, but it is not even dominant, being primarily post-Anselmian and within certain branches of Protestantism.

Of lesser importance:

The title would better use “Rereading” instead of “Relecture.”

Pg. 5, end of first full paragraph: It hasn’t been explained previously what this “anti-God dynamic” is and there’s not even anything in the article up to that point by which the reader may surmise.

Pg. 5, 3rd line of the 2nd full paragraph: “questioned by exegetes” begs for a citation of an example or two.

Pg. 5, section 2.3, line 4: I would suggest “Enlightenment” rather than “enlightened”.  The sentence is intending to refer to an historical approach that is not necessarily actually enlightened (and thus seemingly correct), but an approach that emerged from the Enlightenment.

The paragraph at the bottom of p. 12, top of 13 is a verbatim repeat of the end of the previous paragraph.

Typos and errors are scattered throughout (e.g., “fife” instead of “five”, “ist” instead of “is”).

Author Response

Dear reviewer,
Thank you for the encouraging feedback for my project. 
You object: „Specifically, the application of Schwager‘s/Anselm‘s method to engaging in „repeated new readings“ does not seem to belong in an article that is centered on developing a cairological dramatic theology of divine action.“ 
This methodological approach is necessary for me in order not to lose the advantages of Schwager's dramatic Christology in its further development. These advantages seem to me essential for a theology of God's action with the aims mentioned at the beginning (in view of the tension between biblical and today widespread faith in God's saving power and in view of a polarisation in relation to charismatic and evangelical renewal movements). What I need them for will become clearer in the second essay, in view of the most important objection to a theology that gives much credit to God's saving power: that of theodicy (cf. my references to Stosch).
I can agree with you when you write: „I think that the two can be woven together coherently within this work, but currently that is not made clear“, and also your point: „As the article progresses, the reader needs a couple reminders of the main point of the article“. 
For this purpose, I have now inserted a section in the middle of the essay, at the beginning of Chapter 5, which brings together the connection between the two disparate parts (God's action and methodology) and clarifies in advance what I intend to do with the readings that will now begin. I have also rewritten the abstract so that the connection between the two parts becomes clearer.
Regarding your criticism that the article contains too few references to different theologies of God's action: The conceptualisation of my project in two essays implies that in the first part I develop the basic lines of a new approach to God's action in order to bring them into a discussion with other approaches only in the second part. I also point this out clearly in the essay. This is also the reason why I consider only few other approaches in the first part. In the final chapter, however, I have taken at least two important approaches to God's action into account: in addition to at least a reference to Reinhold Bernhardt's "Was heißt Handeln Gottes", I have gone into more detail with Hans Kessler's typology of God's action in the newly written footnote 35. For English literature on God's action, I need to familiarise myself. I will take this into account in my essay below.
Your references to the question of God's unconditional or presuppositionless initiative of grace in the act of salvation through Jesus Christ are comprehensible to me. I had already written another Relecture chapter on the second act, in which I argue more strongly than Schwager allows for an active action of God in the judgment processes and also on the way to the cross. Here I could have responded to the main objection raised in Marshall's important review of Jesus in the Drama of Salvation. But I had to take this part out because it would have gone beyond the scope of this essay. I hope that I will be able to deal with it in another contribution.
Your suggestions for corrections were also helpful, including the content of p. 4 below, all of which I have taken into account.
Many thanks.